# HOIDiNi: Human-Object Interaction through Diffusion Noise Optimization

## Abstract

We present HOIDiNi, a text-driven diffusion framework for synthesizing realistic and plausible human-object interaction (HOI). HOI generation is extremely challenging since it induces strict contact accuracies alongside a diverse motion manifold. While current literature trades off between realistic motions and accurate contacts, HOIDiNi optimizes directly in the noise space of a pretrained diffusion model using Diffusion Noise Optimization (DNO), achieving both. This is made feasible thanks to our observation that the problem can be separated into two phases: an object-centric phase, primarily making discrete choices of hand-object contact locations, and a human-centric phase that refines the full-body motion to realize this blueprint. This structured approach allows for precise hand-object contact without compromising motion naturalness. Quantitative, qualitative, and subjective evaluations on the GRAB and OMOMO datasets clearly indicate HOIDiNi outperforms prior works and baselines in contact accuracy, visual plausibility, and overall quality. Our results demonstrate the ability to generate complex, controllable interactions, including grasping, placing, and full-body coordination, driven solely by textual prompts. Please watch the supplementary video.

## 1 Introduction

Human-object interaction (HOI) lies at the core of many everyday tasks, such as frying an egg or drinking a cup of water, with crucial applications to any digital human-like agent. Even though it is ubiquitous, this central capability still eludes modern motion generation modeling techniques. This is because HOI modeling requires millimeters-level accuracy to avoid noticeable artifacts, but tackles a diverse motion space, rich with nuanced human behavior.

Indeed, when applying traditional generation techniques to the rather limited HOI data available, results often exhibit physical artifacts like inter-object penetration, floating, or implausible grasps, even when restricted to hand-only scenarios (Huang et al., 2025; Zhang et al., 2025a; Li et al., 2024a).

To facilitate desired accuracy while maintaining plausibility, most recent literature employs generation guidance (Peng et al., 2023; Diller & Dai, 2024; Zhang et al., 2025b) or post-generation optimization (Wu et al., 2024; Ghosh et al., 2023). While this can improve physical correctness, both of these approaches adhere to fine-grained contact requirements by pulling the motion off the human manifold, on the account of realism.

In this work, we present **H**uman-**O**bject **I**nteraction through **Di**ffusion **Noi**se optimization (HOIDiNi), a text-driven diffusion framework that satisfies the tight constraints of HOI while remaining on the manifold of realistic human motion. We address this challenge using an optimization strategy that, by design, preserves the learned motion distribution: Diffusion Noise Optimization (DNO) (Karunratanakul et al., 2024), a test-time sampling method that traverses the noise space of a pretrained diffusion model to steer generation toward desired losses. Originally applied to control free-form motion synthesis, DNO proves to be a natural fit for HOI when carefully adapted to the structure and demands of the task.

We begin by training a diffusion model, CPHOI, to learn the joint distribution of full-body human motion and object trajectories, enabling coordinated interaction within a unified generative space. A key insight is that accurate HOI depends on identifying semantically meaningful contact pairs

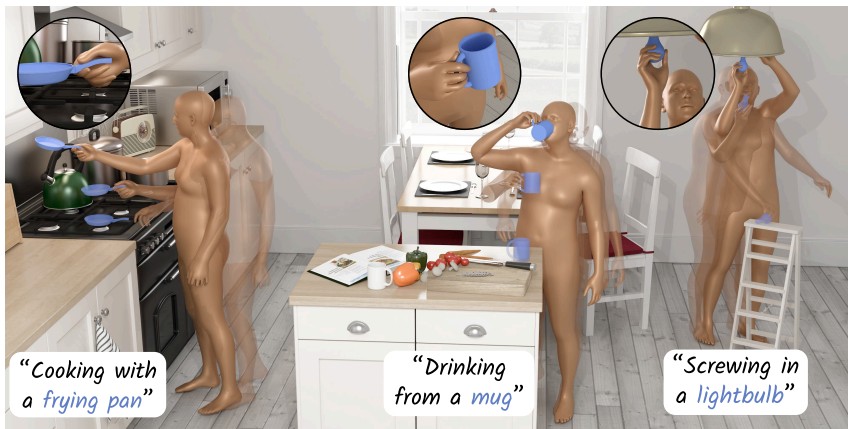

Figure 1: HOIDiNi generates human-object interactions from text descriptions and object geometry, integrated here into a 3D scene from Jay-Artist (2012).

between the palm surface and the input object's surface. Unlike prior methods that rely on heuristics, CPHOI dynamically predicts these contacts for each frame in addition to full-body, fingers, and object trajectories, allowing precise, frame-consistent interaction that adapts to object shape and motion, resulting in more stable and realistic behaviors.

As it turns out, using diffusion noise optimization over this joint discrete/continuous space of Contact-Pairs, Human, and Object motions is challenging, with many local discontinuities that destabilize convergence. We observe that the complexity of HOI optimization can be separated into two optimization phases. The first, Object-Centric phase considers the motion of the object and its contacts with the hands only, forming a reliable structural blueprint for the ensuing full-body motion. This outline then guides the second, Human-Centric phase, which completes the full-body motion, refining finger articulation for precise grasping, and generating natural body posture that semantically supports the object's behavior and dynamics.

A central challenge in the first phase is determining which locations on the hands should make contact with the object and where. Typically in prior works Zhang et al. (2025b; 2024a), this is done using nearest-neighbor heuristics, but this approach is brittle, especially for small or thin objects, and highly sensitive to initialization. Instead, we explicitly predict contact pairs between the hand and object surfaces, and optimize them jointly with the object's 6-DoF trajectory. The DNO objective enforces semantically meaningful placement while preventing interpenetration with supporting surfaces.

After this outline of object motion and contact locations is determined and fixed, the second phase then optimizes the full-body motion, including the fingers, conditioned on the object trajectory and contact pairs. In phase case, DNO helps satisfy these contacts without penetrating the object. Throughout, the DNO process keeps the samples close to the motion manifold, ensuring realism.

Quantitative evaluations on the GRAB (Taheri et al., 2020b) and OMOMO (Li et al., 2023) datasets demonstrate that HOIDiNi outperforms prior baselines in both interaction accuracy and motion realism, as measured by contact precision, physical validity, and proximity to the human motion manifold. Subjectively, a user study indicates dramatic preference to our resulting motions compared to competing literature and baselines. Qualitatively, we showcase a range of full-scene interactions, including object grasping and placement, all driven by textual instructions, trained on a single dataset. These results highlight HOIDiNi's ability to synthesize complex, controllable, and visually plausible HOI behaviors.

## 2    RELATED WORK

**Controlled Motion Synthesis.** Current motion synthesis methods increasingly focus on controllability. TEMOS (Petrovich et al., 2022) and MotionCLIP (Tevet et al., 2022) addressed text-to-motion synthesis using a Transformer VAE (Vaswani et al., 2017; Kingma & Welling, 2013). T2M (Guo et al., 2022b), T2M-GPT (Zhang et al., 2023), and MoMask (Guo et al., 2024) adopt a VQ-VAE (Van Den Oord et al., 2017) to quantize motion and generate it sequentially in the latent space, conditioned

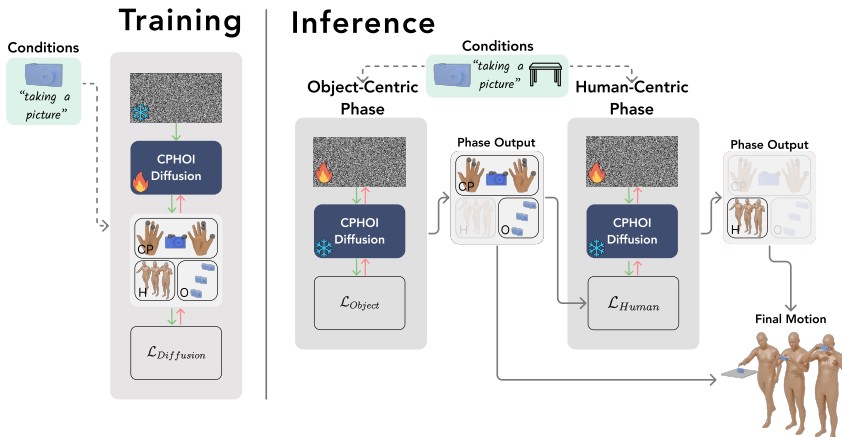

Figure 2: **System Overview.** HOIDiNi generates Human-object Interaction (HOI) motions according to a text prompt, a mesh describing the object, and the occupied volume in the scene, by optimizing the diffusion noise. The *Object-Centric Phase* generates the object motion and its contact points with the hands ($CP$ and $O$), then the *Human-Centric Phase* follows and generates the full human motion($H$): body and fingers, adhering to the constraints implied by the previous phase. Both phases use CPHOI, a pre-trained diffusion model that learned the human-object joint distribution. We apply Diffusion Noise Optimization (DNO) (Karunratanakul et al., 2024) to fulfill the two sets of loss functions ($\mathcal{L}_{\text{Object}}$ and $\mathcal{L}_{\text{Human}}$) without deviating from the learned distribution.

on text. MDM (Tevet et al., 2023) and MoFusion (Dabral et al., 2023) introduced the denoising diffusion framework (Ho et al., 2020b) to motion synthesis, demonstrating its effectiveness across multimodal tasks such as action-, text-, and music-to-motion (Tseng et al., 2023).

The diffusion paradigm enables diverse control mechanisms: PriorMDM (Shafir et al., 2024), Cond-MDI (Cohan et al., 2024), GMD (Karunratanakul et al., 2023), and OmniControl (Xie et al., 2023) combine temporal conditioning with Classifier-guidance (Dhariwal & Nichol, 2021) to achieve joint-level control. MoMo (Raab et al., 2024) demonstrated motion transfer through attention injection, while LoRA-MDM (Sawdayee et al., 2025) employed Low-Rank Adaptation (Hu et al., 2022) for motion stylization. CAMDM (Chen et al., 2024) and A-MDM (Shi et al., 2024) accelerated sampling by introducing autoregressive motion diffusion. CLoSD (Tevet et al., 2025) further integrated autoregressive diffusion into a physics-based simulation framework for object interaction.

Diffusion Noise Optimization (DNO) (Karunratanakul et al., 2024; Ben-Hamu et al., 2024) proposes applying spatial constraints by optimizing the initial diffusion noise, enabling precise free-form control. DartControl (Zhao et al., 2025) built on this idea by accelerating the process through autoregressive diffusion. We show that DNO can be extended to the millimetric accuracy required for object interaction, and introduce a two-phase DNO strategy tailored for object interactions.

**Human-Object Interaction.** Early HOI methods generate motions in stages: SAGA (Wu et al., 2022) first predicts a static target frame, then interpolates motion via a VAE decoder; GOAL (Taheri et al., 2022) adds optimization to align motion and object; IMoS (Ghosh et al., 2023) uses dual-stream autoregressive networks for arm and body motions followed by object alignment; TOHO (Li et al., 2024b) predicts the object's final position, generates grasping poses, and fills the trajectory with an implicit representation. Diffusion-based methods have recently gained traction: OOD-HOI (Zhang et al., 2024b) uses a dual-branch reciprocal diffusion model with IMoS-style refinement; HOI-Diff (Peng et al., 2023), CHOIS (Li et al., 2024a), and DiffGrasp (Zhang et al., 2025b) apply classifier guidance, with HOI-Diff using affordances and CHOIS and DiffGrasp goal functions; OMOMO (Li et al., 2023) models hand-object paths first, then full-body motion; BimArt (Zhang et al., 2024a) conditions contact generation on object trajectories to guide body motion. Both methods reduce task complexity by generating only partial motion (hands or body), enabling more tractable modeling at the cost of full-scene coherence. Finally, CLoSD (Tevet et al., 2025) and Wu et al. (2024) apply physical trackers atop generated motion, improving grasp accuracy at the cost of realism.

Concurrent to this work, CoDa (Pi et al., 2025) applied Diffusion Noise Optimization with separate hand and body models, thus not learning the joint Hand–Body distribution.

## 3 PRELIMINARIES

**Denoising diffusion models.** Denoising diffusion probabilistic models (DDPMs) (Ho et al., 2020a) define a forward Markov process $\{x_t\}_{t=0}^{T}$ that progressively adds Gaussian noise to a data sample $x_0 \sim p_{\text{data}}(x_0)$: $q(x_t \mid x_{t-1}) = \mathcal{N}(\sqrt{\alpha_t}x_{t-1}, (1 - \alpha_t)I)$, with $\alpha_t \in (0, 1)$. As $t$ increases, $x_T$ approaches $\mathcal{N}(0, I)$. The reverse process learns to denoise back to $x_0$, optionally conditioned on $c$ (e.g., text or pose). Unlike the original DDPM that predicts noise $\epsilon_t$, we follow MDM (Tevet et al., 2023) and predict $\hat{x}_0$ directly, yielding the training objective: $\mathcal{L}_{\text{simple}} = \mathbb{E}_{x_0 \sim p(x_0|c),\, t \sim [1,T]} \left[ |x_0 - \hat{x}_0|_2^2 \right]$.

**Diffusion noise optimization (DNO).** A common approach to constrain a data sample $x \sim X$ is to directly optimize it via $x^* = \arg\min_x \mathcal{L}(x)$, where $\mathcal{L}$ encodes task-specific objectives. In the context of HOI, such post-hoc optimization has been widely used (Ghosh et al., 2023; Zhang et al., 2024a; Paschalidis et al., 2024), but it lacks guarantees that $x^*$ remains within the data distribution $X$, often resulting in unrealistic outputs. A more robust strategy is to optimize in a latent space $z \sim Z$, assuming $x = D(z)$ for some decoder $D$. The optimization becomes:

$$z^* = \arg\min_z \{\mathcal{L}(D(z)) + \mathcal{R}(z)\}$$

where $\mathcal{R}(z)$ encourages $z \sim Z$, thereby keeping the final output $x^* = D(z^*)$ on-manifold. This technique is commonly employed with VAEs (Holden et al., 2016; Pavlakos et al., 2019b) and GANs (Karras et al., 2020b; Patashnik et al., 2021) across both image and 3D domains.

DNO (Karunratanakul et al., 2024; Ben-Hamu et al., 2024) extends this principle to diffusion models, treating the latent variable as the initial noise $x_T \sim \mathcal{N}(0, I)$, and the decoder as the full sampling process of a pretrained diffusion model $G$, resolved using the ODE formulation of DDIM (Song et al., 2021). The optimization is then defined over $x_T$, with gradients propagated through all denoising steps:

$$x_T^* = \arg\min_{x_T} \{\mathcal{L}(\text{ODE}(G, x_T)) + \mathcal{R}_{\text{decorr}}(x_T)\}$$

Where the final output will be $x^* = \text{ODE}(G, x_T^*)$. Here, $\mathcal{R}_{\text{decorr}}$ is a decorrelation regularizer (Karras et al., 2020a) that encourages $x_T$ to remain within the Gaussian prior. HOIDiNi extends this formulation with a two-phase DNO strategy tailored for HOI, enabling precise contact control while preserving motion realism.

## 4 METHOD

An overview of HOIDiNi is illustrated in Figure 2. Our goal is to generate realistic, contact-rich human-object interactions (HOI) by guiding a diffusion model to satisfy task-specific constraints without drifting off the motion manifold.

We begin by defining a structured data representation that jointly encodes full-body human motion, object trajectories, and accurate surface-level contact points (4.1). Then, we turn to describe CPHOI, a diffusion model that captures the joint distribution of human and object motion along with dense contact predictions (4.2). Our model predicts contact correspondences directly, which proves essential for stable and semantically meaningful grasps.

At inference, we employ a two-phase Diffusion Noise Optimization (DNO) (Karunratanakul et al., 2024) strategy tailored to HOI (4.3). The first, object-centric, phase optimizes the object's trajectory and contact pairs based on scene constraints such as placement and support. The second, human-centric, phase completes the full-body motion, including hand articulation, to fulfill the previously determined contact goals while maintaining realistic posture and avoiding collisions. This structure allows us to satisfy complex physical constraints while remaining within the learned motion distribution.

### 4.1 DATA REPRESENTATION

Our diffusion model, CPHOI, generates triplets of the form $(CP, H, O)$, representing Contact Pairs, Human motion, and Object motion, respectively. We will now elaborate on each of these components.

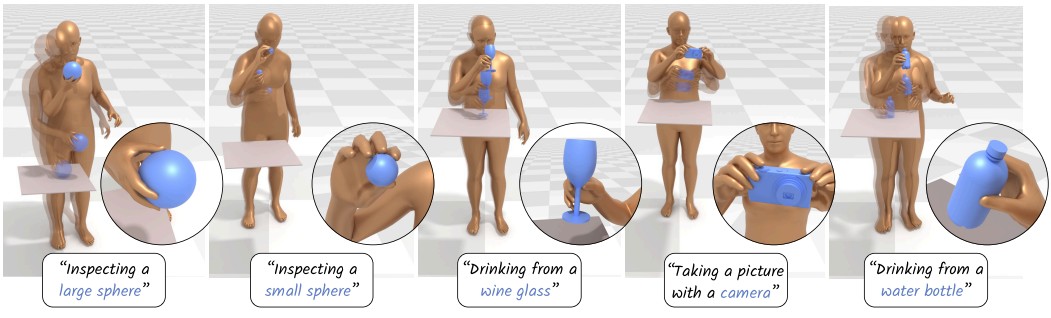

"*Inspecting a large sphere*"    "*Inspecting a small sphere*"    "*Drinking from a wine glass*"    "*Taking a picture with a camera*"    "*Drinking from a water bottle*"

Figure 3: **Qualitative Results** of human-object interactions generated by our method across diverse prompts. For instance, "taking a picture with a camera" yields a semantically appropriate two-handed pose. Motions are both visually plausible and aligned with the prompts.

**Contact Pairs Sequence.** The Contact Pairs sequence is central to our method, generated during the first, object-centric, phase and serve as conditioning for the second phase. Unlike previous approaches, that snap the given or predicted object trajectory to the hands using simple heuristics, we adopt a learned approach to semantically predict these contacts. We define a discrete set of Anchor points $\mathcal{A}$ on the fingers and palms, denoting potential contacts between the hands and the object (See Appendix). At each frame $f$, and for each anchor $a \in \mathcal{A}$, we represent and predict contacts using a binary variable $b_a$, indicating whether the anchor is in contact, and a corresponding position $p_a \in \mathbb{R}^3$, specifying the contact location on the object surface in its rest pose. This yields the **contact representation:** $F_{CP} = [p_1, \ldots, p_{|\mathcal{A}|}, b_1, \ldots, b_{|\mathcal{A}|}]$ of dimension $(3+1) \times |\mathcal{A}|$ per-frame in the sequence.

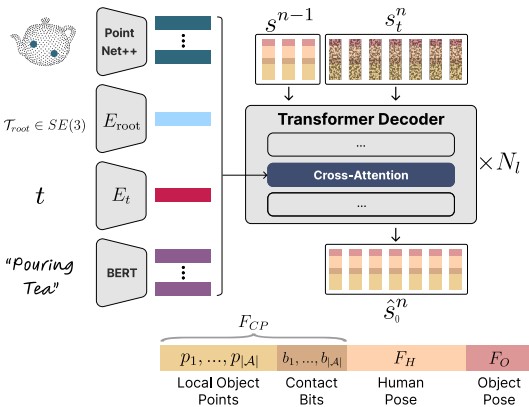

Figure 4: **CPHOI Diffusion Model.** CPHOI autoregressively predicts the next motion segment $s^n$ from the previous one $s^{n-1}$. The figure illustrates a single diffusion step, where the model denoises $s_t^n$ to predict $\hat{s}_0^n$. It jointly generates human and object motions, along with dynamic contact points, conditioned on the object's geometry and a text description of the interaction.

**Human Motion Representation.** We adopt a variant of the widely used HumanML3D (Guo et al., 2022a) representation to encode human motion. The per-frame human feature is defined as: $F_H = [r_z^H, \dot{r}_x^H, \dot{r}_y^H, \dot{\alpha}^H, \theta^H, j^H]$ where $r_z^H$ is the vertical root height, $(\dot{r}_x^H, \dot{r}_y^H)$ denotes the planar root velocity, $\dot{\alpha}^H$ is the angular root velocity, $\theta^H$ contains the SMPL-X pose parameters, and $j^H$ denotes the relative 3D joint positions.

The 52 relevant joints from the SMPL-X model are used, including both body and hand joint rotations (but not shape). Unlike the original representation, we directly employ the SMPL-X pose parameters $\theta^H$, allowing us to extract the human mesh in a fully differentiable manner. This property is essential for enabling backpropagation during the diffusion noise optimization process.

**Object Motion Representation.** The per-frame object's pose is, $F_O = [\theta^O, r^O, \dot{r}^O]$, where $\theta^O$ denotes the object's global rotation represented in Cont6d format, $r^O$ is the object's global translation, and $\dot{r}^O$ is its linear velocity. Together, these parameters define the object's 6DoF trajectory.

**Final representation.** Combining the representations of our three data components, we end up with a feature representation for each frame with the form $\mathbf{F} = [F_{CP}, F_H, F_O]$ (Figure 4, bottom)

## 4.2 CPHOI DIFFUSION MODEL

Our model, CPHOI, is illustrated in Figure 4. To support DNO-based optimization, which repeatedly queries the generative model and is therefore computationally demanding, we require a fast and

efficient architecture. We design CPHOI as a lightweight, text-driven, autoregressive diffusion model that incorporates geometric understanding of the object and interaction semantics. Autoregressive diffusion, as shown in prior work (Shi et al., 2024; Chen et al., 2024), is significantly faster than full-sequence denoising. It processes shorter segments per step and requires fewer diffusion iterations. This efficiency is crucial for accelerating DNO (Zhao et al., 2025). Our model is inspired by the autregressive design of DiP (Tevet et al., 2025) which also enables high-level control descriptions as those needed for our task.

CPHOI generates the next motion segment $s^n = [F_i]_{i=1}^L$ of $L$ frames, given the previous segment, $s^{n-1}$, the object geometry in point-cloud format, and a text prompt that describes the interaction. For each denoising step $t \in [0, T]$, the inputs to the transformer decoder backbone are the frames of the previous segment $s^{n-1}$, followed by the current segment to be denoised, $s_t^n$. The model predicts the clean version of the segment, $\hat{s}_0^n$.

The object's geometry, the text condition, and the timestep $t$ are encoded into a sequence of latent embeddings and injected through the cross-attention of each transformer layer, at each denoising step. To encode the *geometry of the interacted object*, we use a shape encoder with a PointNet++ (Qi et al., 2017) architecture, which is trained simultaneously with the diffusion model. We uniformly sample the mesh using $V$ points, and encode each one into a latent descriptor of length $C$ ($V = 512$ and $C = 512$ in all our experiments). The *text prompt* is encoded into a sequence of embeddings using a pre-trained and fixed DistilBERT (Sanh, 2019) model. The *diffusion timestep* $t$ is embedded using a standard positional embedding denoted $E_t$.

Since object location is represented in global coordinates, whereas the human is relative to the previous frame, we inform the model regarding the global position of the body. We hence encode the global root transformation $\mathcal{T}_{root} \in SE(3)$ at the first frame of the autoregressive sequence.

Finally, all tokens belonging to the same signal type are augmented with a learned type-specific embedding, allowing the model to distinguish between the different sources of information during cross-attention.

### 4.3 Two-phase Generation Optimization

Accurate body-object contacts are a key component of plausible HOI motion, but represent a discontinuous space, with discrete and unstable decision making. Thus we found the straightforward optimization, using a single step challenging in practice (see Figure 7). To address this, we separate the process into two phases, each solving a different part of the problem variables:

**Phase 1: Object-Centric.** In this phase, we optimize the object related part only to outline the motion, based on the given prompt, object geometry, and scene constraints (e.g., table surface). Although the full output of the model is the triplet $(CP, H, O)$—representing the contact-pair sequence, human motion, and object motion respectively—only $(CP)$ and $(O)$ are considered in this stage. We found weight sharing between phases to improve performance. The objective $\mathcal{L}_{Object}$ for this phase is defined to avoid object-scene penetrations and optionally to guide the object toward a set of target poses in specified frames (as sometimes used in the literature (Li et al., 2024a)):

$$\mathcal{L}_{Object} = \lambda_{POS}\mathcal{L}_{POS} + \lambda_{Goal}\mathcal{L}_{Goal} + \lambda_{Static}\mathcal{L}_{Static}$$

This objective comprises a penetration term between the object and the scene $\mathcal{L}_{POS}$, a goal term encouraging the object to achieve specified poses in specified times $\mathcal{L}_{Goal}$, and the static term, $\mathcal{L}_{Static}$, that encourage the object to stay static in frames without predicted contact. The last two are detailed in Appendix C.

**Penetration Loss.** To measure interpenetration between two watertight meshes, denoted $M_a$ and $M_b$, we define a bidirectional penetration loss that penalizes vertices of one mesh that are located inside the volume enclosed by the other. Each mesh $M$ consists of a set of vertices $\mathcal{V}$ and a set of polygons $\mathcal{F}$. Each polygon $f \in \mathcal{F}$ is associated with an outward-facing normal vector $\mathbf{n}_f$.

For each vertex $\mathbf{v} \in \mathcal{V}_a$, we identify whether it is inside the other mesh simply through randomly casting a single ray, and checking the normal direction of the intersected triangle if it exists. We denote the vertices found as inside $\mathcal{V}_a^{in} \subset \mathcal{V}_a$.

Figure 5: **Comparisons.** HOIDiNi generates semantically correct and accurate interaction with the gaming controller. From left to right: IMoS (Ghosh et al., 2023) optimization yields inferior contacts and unrealistic motion; CPHOI *inference only* generates decent poses but fails to satisfy contacts; Our losses with *Classifier Guidance* brings the object insufficiently closer. Replacing our contact-point prediction with the popular *nearest-neighbor* heuristic fails to choose correct contacts, in contrast to our plausible and human-like result.

For every vertex $v \in \mathcal{V}_a^{\text{in}}$, we project it on $M_b$, and denote the projection distance as $\text{NN}(\mathbf{v}; M_b)$. We then define the penetration loss from $M_a$ into $M_b$ as:

$$\mathcal{L}_{\text{pen}}(M_a \rightarrow M_b) = \frac{1}{|\mathcal{V}_a|} \sum_{\mathbf{v} \in \mathcal{V}_a^{\text{in}}} \|\mathbf{v} - \text{NN}(\mathbf{v}; M_b)\|^2$$

Hence, the full symmetric loss is, $\mathcal{L}_{\text{Penetration}} = \mathcal{L}_{\text{pen}}(M_a \rightarrow M_b) + \mathcal{L}_{\text{pen}}(M_b \rightarrow M_a)$.

**Phase 2: Human-Centric Phase.** This phase refines the human motion to conform to the fixed object motion and contact sequence $(CP, O)$ produced in Phase 1. With the contact points fixed, this optimization over the human manifold is much more stable. The objective of this diffusion-based optimization is defined as:

$$\mathcal{L}_{Human} = \lambda_C \mathcal{L}_C + \mathcal{L}_{P_H} + \lambda_{Foot} \mathcal{L}_{Foot} + \lambda_{Jitter} \mathcal{L}_{Jitter}$$

where $\mathcal{L}_C$ denotes the contact loss, promoting alignment of the human body with predefined contact points on the object. The human penetration loss term $\mathcal{L}_{P_H}$ combines three components:

$$\mathcal{L}_{P_H} = \lambda_{P_{HO}} \mathcal{L}_{P_{HO}} + \lambda_{P_{HS}} \mathcal{L}_{P_{HS}} + \lambda_{P_{HH}} \mathcal{L}_{P_{HH}},$$

representing human-object, human-scene, and human-human penetration losses, respectively. Specifically, $\mathcal{L}_{P_{HO}}$ penalizes interpenetration between the human and the object, $\mathcal{L}_{P_{HS}}$ prevents collisions with the static scene, and $\mathcal{L}_{P_{HH}}$ reduces self-intersections within the human mesh, particularly between the hands. The contact loss $\mathcal{L}_C$ uses $L_2$ distance to push hand vertex locations to targets on the moving object, as dictated by $CP$ and $O$ from the previous phase. The penetration loss terms are similar to the penetration term $\mathcal{L}_{P_{OS}}$ defined in phase 1. Together, these losses represent physical plausibility, encouraging surface contacts while avoiding penetration, and the optimization scheme ensures realism and human likeness.

## 5 EXPERIMENTS

### 5.1 EVALUATION SETTING

**Implementation Details.** Our code and checkpoints will be made available; Please watch the supplementary video. CPHOI is implemented as a Transformer decoder architecture with 8 layers and a hidden dimension of 512. Our point-wise object embedding network is a PointNet++ (Qi et al., 2017) fed with 512 randomly sampled vertices from the conditioned object. We condition the motion on a prefix of 15 frames and generate 100 frames. The model is trained with DDPM (Ho et al., 2020b); DDIM (Song et al., 2020) is used at inference. Further details are at Appendix A.

| Experiment | FID ↓ | Diversity → | AVE ↓ | IRA ↑ | Multimodality → | Penetration (mm) ↓ | Floating (mm) ↓ |
|---|---|---|---|---|---|---|---|
| GT | – | 0.995 | – | 74.6% | 0.194 | 5.0 ± 1.8 | 2.6 ± 1.8 |
| IMoS | 0.205 | **1.026** | 0.121 | 46.5% | 0.204 | **3.0 ± 7.7** | 52.2 ± 53.8 |
| HOIDiNi (Ours) | **0.159** | 0.996 | **0.121** | 62.3% | 0.245 | 6.8 ± 2.9 | **2.3 ± 4.7** |
| Inference Only | **0.144** | 1.054 | 0.145 | **68.4%** | **0.214** | 16.1 ± 20.9 | 151.4 ± 108.6 |
| Single-Phase | 0.221 | 1.026 | 0.140 | 43.0% | 0.241 | 7.8 ± 3.6 | 20.1 ± 21.7 |
| Phase1 Inference, Phase2 DNO | 0.148 | 0.923 | **0.172** | 65.8% | 0.217 | 6.8 ± 2.2 | 2.0 ± 4.1 |
| Classifier Guidance | 0.149 | **1.013** | 0.149 | **68.4%** | 0.215 | 14.4 ± 20.5 | 166.8 ± 143.2 |
| Higher Penetration Coef. $\lambda_{P_{HO}}$ | 0.156 | 1.015 | 0.144 | 57.0% | 0.225 | **5.6 ± 2.9** | 5.9 ± 7.2 |
| NN instead of Contact Pairs | 0.162 | 1.040 | 0.153 | 57.9% | 0.245 | 7.7 ± 3.8 | 4.3 ± 5.0 |
| No Jitter Loss | 0.160 | 1.026 | 0.143 | 60.5% | 0.220 | 7.1 ± 2.7 | **1.9 ± 4.0** |

Table 1: **Quantitative Results and Ablation Study.** Comparison on the GRAB dataset (Taheri et al., 2020b) using IMoS-defined (Ghosh et al., 2023) metrics for motion realism, along with average floating and penetration errors (in millimeters). → indicates that better is closer to ground-truth performance.

**Data.** We evaluate our approach using the GRAB (Taheri et al., 2020a) and OMOMO (Li et al., 2023) datasets. **GRAB** includes SMPL-X (Pavlakos et al., 2019a) human motion parameters, object 6DoF trajectories, and per-vertex contact force data for both human and object meshes. The dataset comprises 1,334 motion samples from 10 different subjects. Since different human subjects in the dataset exhibit varying shape parameters $\beta$, and considering the dataset's relatively small scale compared to datasets like HumanML3D, we re-target all motions to a standardized, neutral SMPL-X model with shape parameters fixed to $\beta = 0$. Additionally, we used ChatGPT 4o to enhance its discrete action annotations into text prompts. The **OMOMO** dataset includes 27,952 motion

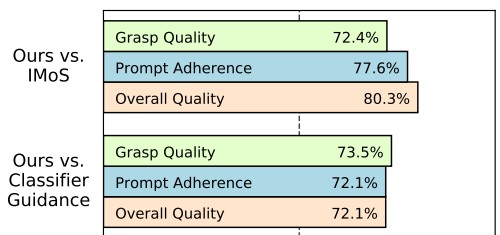

Figure 6: **User Study.** We compare to two baselines, measuring on grasp quality, prompt adherence, and overall quality over 12 random samples, each evaluated by at least 10 users. The dashed line marks 50% ratio.

sequences, interacting with 13 objects. It **does not include finger motions** and hence doesn't enable precise interactions. We use it strictly for comparisons with the recent literature (Li et al., 2024a). Following it, we use the data's subset, called FullBodyManipulation.

**Baselines** We compare our approach against IMoS (Ghosh et al., 2023), CHOIS (Li et al., 2024a), and several internal baselines. One baseline applies our model at inference time without any optimization. Another replaces the diffusion noise optimization with classifier guidance. The third substitutes the predicted contact pairs with nearest-neighbor-based pairs. To enable a fair comparison with classifier guidance, we adopt DDPM hyper-sampling and increase the number of denoising steps from 8 to 500, consistent with Peng et al. (Peng et al., 2023). We report results using the optimal guidance scale. For the nearest-neighbor baseline, contact assignment is updated at each optimization step based on the current nearest neighbors. Additionally, we explore increasing the penetration loss weight by setting $\lambda_{P_{HO}} = 0.9$.

**Metrics.** We evaluated our method on motion realism and interaction accuracy. For GRAB, we followed IMoS (Ghosh et al., 2023) metrics, computing *FID*, *Diversity*, *Multimodality*, *Intent Recognition Accuracy (IRA)*, and *Average Variance Error (AVE)* using a classifier trained on full-body joint positions, hand joints, and object trajectories (Appendix D). Grasp accuracy was measured via two failure modes: *Penetration* (mean depth in mm, frames with penetration only) and *Floating* (frames without penetration). For OMOMO, we used the CHOIS benchmark, covering condition adherence, motion fidelity, and interaction accuracy (Appendix D).

## 5.2 Results

Table 1 compares our method to IMoS. HOIDiNi achieves better FID and IRA, indicating improved realism, and significantly reduces floating while maintaining comparable penetration levels. Table 2

| Method | Condition Matching | | | Human Motion | | | | Interaction | | | | |
|---|---|---|---|---|---|---|---|---|---|---|---|---|
| | $T_s \downarrow$ | $T_e \downarrow$ | $T_{xy} \downarrow$ | $H_{\text{feet}} \downarrow$ | FS $\downarrow$ | $R_{\text{prec}} \uparrow$ | FID $\downarrow$ | $C_{\text{prec}} \uparrow$ | $C_{\text{rec}} \uparrow$ | $C_{F_1} \uparrow$ | $C_\%$ | $P_{\text{hand}} \downarrow$ |
| CHOIS | 1.90 | 6.90 | 2.81 | 4.48 | 0.34 | **0.43** | **0.97** | **0.80** | 0.64 | 0.67 | 0.56 | **0.61** |
| HOIDiNi (ours) | **0.00** | **0.00** | **0.00** | **3.17** | **0.30** | 0.42 | 1.24 | 0.78 | **0.8** | **0.77** | 0.76 | 0.67 |

Table 2: Comparison to CHOIS (Li et al., 2024a) over the OMOMO (Li et al., 2023) dataset. Measuring condition matching, human motion, and interaction via a set of metrics defined by CHOIS.

compare our model to CHOIS using the metrics defined by them and shows that HOIDiNi improves the contact accuracy while maintaining comparable motion fidelity. The *condition matching* metrics demonstrate the preciseness of the DNO mechanism, which consistently delivers zero error.

**User Study.** We conducted a user study for the GRAB dataset with 24 participants, evaluating 12 side-by-side randomly selected samples of two models using the same inputs. As shown in Figure 6, users preferred the results generated by our framework. A representative screenshot from the study interface is shown in Figure 8.

**Ablation Study.** Table 1 summarizes our ablations. Inference-only results stay on-manifold but suffer from severe penetration and floating. Omitting object-centric optimization shows that most constraints are resolved in the human-centric phase. Using nearest-neighbor contacts instead of predicted ones significantly harms FID and IRA, indicating reduced realism.

**The two-phase design** is investigated in Figure 7, comparing to the single-phase setup, where contact predictions are optimized jointly with human motion. The single-phase approach results in frequent updates to contact assignments throughout the process. These updates shift the contact loss objective over time, making the optimization less stable and harder to

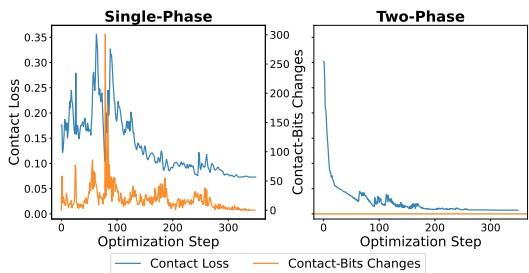

Figure 7: **Single vs. Two-Phase Optimization.** Comparison of contact loss and changes in predicted contact bits during DNO optimization (shared y-axis). **(Left):** In the single-phase setup, contact predictions evolve alongside motion, causing unstable objectives and hindering convergence. **(Right):** In our two-phase approach, contact pairs are fixed after the object-centric phase, resulting in stable contact loss during the human-centric phase presented here. In this example, the two-phase setting converges to a 10× lower value. *Contact-Bits Changes* denote the number of bit flips in the predicted contact matrix between successive steps.

converge. In contrast, our two-phase approach separates contact prediction and motion optimization: contact pairs are predicted and fixed in the first (object-centric) phase, allowing the second (human-centric) phase to optimize a stable and well-defined contact loss objective. This decoupling enables more consistent optimization behavior and improves convergence, as reflected in Table 1.

**Qualitative Results.** The supplementary video showcases a variety of motions generated by our model, along with visual comparisons to baseline methods. Figure 10 shows contact pairs generated by HOIDiNi. Figure 5 presents an interaction generated by HOIDiNi for the prompt "The person is playing with the gaming controller." The IMoS baseline demonstrates a common failure where contacts are lacking semantic meaning and visually plausibility due to the fixed-contact snapping approach. Similarly, only replacing our predicted contact pairs with nearest-neighbor assignments also results in incorrect contacts and implausible motion. We further witness that applying our losses through classifier guidance improves hand-object proximity compared to inference-only, but does not produce realistic interactions. Figure 3 presents additional examples generated by our method. These results further highlight the semantic correctness and visually plausibility of the synthesized motions across a diverse set of prompts and object types.

## 6 CONCLUSIONS

We introduced HOIDiNi, an approach for high-precision Human–object Interaction that provides motion fidelity through Diffusion Noise Optimization. Our results demonstrate accurate contact handling and natural motion across complex interaction scenarios. Beyond HOI, we view HOIDiNi as a platform for cases where high-precision is required and regular diffusion inference fails. We encourage the community to use HOIDiNi to advance controllable, high-fidelity motion generation.

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

# APPENDIX

## A  IMPLEMENTATION DETAILS

CPHOI is implemented as a Transformer decoder architecture with 8 layers and a hidden dimension of 512. Our point-wise object embedding network is a PointNet++ (Qi et al., 2017) fed with 512 randomly sampled vertices from the conditioned object. The model is trained with DDPM (Ho et al., 2020b); DDIM (Song et al., 2020) is used at inference. More details per experiment can be found in Table 3. We use the Adam optimizer (Kingma, 2014) for the noise optimization procedure. To reduce memory costs, we focus only on the MANO (Romero et al., 2017a) subset of vertices of the SMPL-X human body and simplify it further to 1,100 vertices for each hand. The object meshes are simplified to 3,000 faces.

| | Experiment | GRAB [2020a] | OMOMO [2023] |
|---|---|---|---|
| Training parameters | # input prefix frames | 15 | 1 |
| | # generated frames | 100 | 119 |
| | diffusion steps ($T$) | 8 | 14 |
| | training steps | 120K | 50K |
| | batch size | 64 | 64 |
| DNO parameters | perturbation scale | $10^{-6}$ | $10^{-5}$ |
| | difference penalty | $10^{-6}$ | $10^{-5}$ |
| | $\lambda_C$ | 0.95 | 0.95 |
| | $\lambda_{Foot}$ | 0.5 | 0.5 |
| | $\lambda_{Jitter}$ | $10^{-5}$ | $10^{-3}$ |
| | $\lambda_{P_{HO}}$ | 0.05 | 0.05 |
| | $\lambda_{P_{HS}}$ | 0.2 | – |
| | $\lambda_{P_{HH}}$ | 0.05 | 0.05 |
| | $\lambda_{P_{OS}}$ | 1.2 | 0.05 |
| | $\lambda_{Goal}$ | 0.5 | 0.9 |
| | $\lambda_{Static}$ | 0.9 | 0.05 |
| | $\lambda_{FeetFloorContact}$ | – | 0.5 |

Table 3: Hyper-parameters in use for each experiment.

## B    CONTACT REPRESENTATION

Directly generating index pairs via diffusion is challenging. To address this, we adopt a more learnable representation. We define a fixed anchor set $\mathcal{A}$, consisting of a subset of MANO (Romero et al., 2017b) hand vertices located on the palm (Figure 9). At each frame $f$, and for each anchor $a \in \mathcal{A}$, a binary variable $b_a$ indicates whether the anchor is in contact. A corresponding position $p_a \in \mathbb{R}^3$ specifies the contact location on the object surface in its rest pose. This yields the contact representation:

$$F_{CP} = \big[p_1, \ldots, p_{|\mathcal{A}|}, b_1, \ldots, b_{|\mathcal{A}|}\big]$$

resulting in a per-frame contact feature of dimension $(3 + 1) \times |\mathcal{A}|$. Figure 10 shows an example of contact pairs generated by HOIDiNi.

For the OMOMO (Li et al., 2023) dataset, which lacks fingers' motion, we follow CHOIS and define the middle finger in each hand in the SMPL-X body model as the anchor, resulting in only two anchors for this benchmark.

## C    ADDITIONAL DNO LOSES

### C.1    OBJECT-CENTRIC LOSSES

**Goal Loss.**    We encourage the object to reach a set of target poses at selected keyframes by penalizing both position and orientation errors. Concretely, we define

$$\mathcal{L}_{\text{Goal}} = \frac{1}{|\mathcal{T}|} \sum_{t \in \mathcal{T}} \Big( \|\hat{\mathbf{t}}_t - \mathbf{t}_t\|^2 + \mathcal{D}_{\text{rot}}\big(\hat{R}_t, R_t\big) \Big)$$

Where, $\mathcal{D}_{\text{rot}}$ measures the angular deviation between rotation matrices $R_1$ and $R_2$.

**Static Loss.**    To prevent unintended object motion, we penalize changes in position across frames where the object is not in contact with the human. We identify contiguous non-contact intervals $\{\mathcal{T}_s^{\text{nc}}\}_{s=1}^S$, and for each such segment $s$, we anchor the object pose to its value at the first frame, $t_s^{\text{start}}$. The loss is then computed as:

$$\mathcal{L}_{\text{Static}} = \frac{1}{\sum_{s=1}^S |\mathcal{T}_s^{\text{nc}}|} \sum_{s=1}^S \sum_{t \in \mathcal{T}_s^{\text{nc}}} \Big( \|\mathbf{t}_t - \mathbf{t}_{t_s^{\text{start}}}\|^2 + \mathcal{D}_{\text{rot}}\big(R_t, R_{t_s^{\text{start}}}\big) \Big),$$

where $\mathbf{t}_t$ and $R_t$ are the object's translation and rotation at frame $t$, respectively, and $\mathcal{D}_{\text{rot}}$ measures the angular distance between two rotations. This encourages the object to remain static when not actively manipulated.

### C.2 HUMAN-CENTRIC LOSSES

**Feet-floor Contact Loss.** For the OMOMO experiment, following CHOIS, we add a loss term that enforces accurate foot contact at the mesh level.

When reconstructing the human mesh with SMPL-X (Pavlakos et al., 2019a) using predicted root positions, joint rotations, and subject-specific body parameters, the generated feet may occasionally fail to touch the floor. To address this, we add a guidance term that encourages realistic feet-floor contact.

Let $\boldsymbol{J}_l$ and $\boldsymbol{J}_r$ denote the positions of the left and right toe joints. At each frame, the supporting foot is identified by comparing their z-coordinates. We further set a threshold height $h = 0.02$ meters, derived from analyzing foot heights in the ground truth motion. The guidance term is then defined as:

$$L_{\text{FeetFloorContact}} = |\min(\boldsymbol{J}_l^z, \boldsymbol{J}_r^z) - h|_2. \tag{1}$$

This measures the vertical deviation between the lower toe joint and the threshold height $h$.

## D EVALUATION METRICS

For both benchmarks, GRAB and OMOMO, we evaluate our method along two dimensions: motion realism and interaction accuracy.

### D.1 GRAB EVALUATION

For the GRAB dataset (Taheri et al., 2020a) experiment, we follow IMoS Ghosh et al. (2023), and compute realism metrics using embeddings from the final layer of an intent classifier. However, the classifier used in IMoS is limited to body joint positions and cannot capture fine-grained grasp dynamics. In contrast, our evaluation employs a more expressive classifier that takes as input body joints, hand joints, and object trajectories, allowing for a more comprehensive assessment of interaction quality.

**FID.** Fréchet inception distance measures the distance between the distributions of generated and ground-truth motions in a learned embedding space. Lower FID values indicate that the synthetic motion is closer in distribution to real motion data, capturing both realism and diversity.

**Diversity.** Evaluates how varied the generated motions are across different samples for the same input condition (e.g., prompt or object). It is computed as the average pairwise distance between multiple motion samples in the embedding space. A lower difference between ground truth and generated diversity scores suggests that the generated motions effectively capture the observed variability of human movement.

**AVE.** Measures the discrepancy between the variance of joint positions in generated motion and that of ground-truth motion. Specifically, it computes the average $L^2$ difference in per-joint positional variance across time. A lower AVE suggests that the model accurately captures the temporal dynamics and variability of natural human movement, avoiding overly rigid or overly jittery outputs.

**IRA.** Intent recognition accuracy quantifies how well the generated motions conveys the intended interaction or action. It is computed as the classification accuracy of the intent classifier on generated samples. High IRA indicates that the generated motions are semantically meaningful and align with their intended action labels, providing a measure of goal consistency and plausibility.

**Multimodality.** Assesses the model's capacity to produce distinct motions for the same conditioning intent. Unlike diversity, which measures sample variation globally, multimodality focuses on conditional variability by comparing multiple outputs conditioned on the same prompt. This metric is crucial for evaluating whether the model can express different plausible interaction strategies.

**Penetration.** Quantifies physical implausibility by measuring the extent to which the human mesh intersects with the object mesh. We compute the mean maximal penetration depth across frames

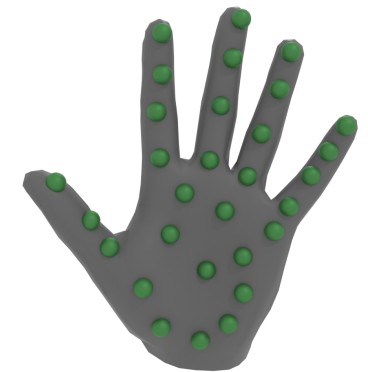

Figure 8: A screenshot from the user study.

Figure 9: Palm anchor set $\mathcal{A}$ used by CPHOI for the GRAB (Taheri et al., 2020a) experiment.

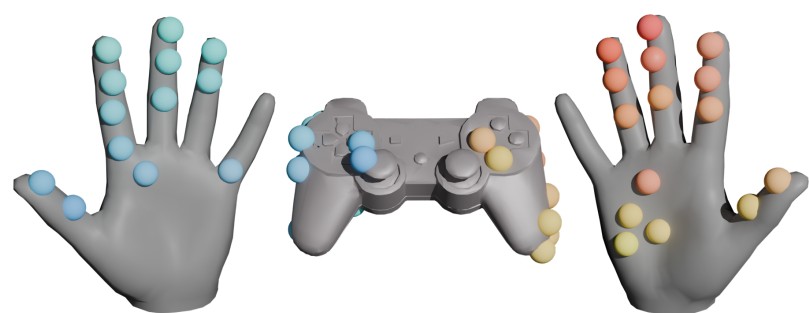

Figure 10: **Contact Pairs.** CPHOI predicts precise, semantically meaningful contact points between the hand and object. Each contact pair is visualized with matching colored spheres.

where interpenetration occurs and the object is above table height. Lower penetration values indicate more physically valid interactions, particularly in grasping and manipulation scenarios where accurate surface contact is essential.

**Floating.** Captures the failure of hand-object interaction where the hand remains unnaturally far from the object surface. It is computed as the mean shortest distance between the body and object meshes, averaged across all motions (excluding frames with penetration and frames where the object is at table height). High floating values typically reflect unrealistic, disconnected grasping motion.

### D.2 OMOMO EVALUATION

For the OMOMO dataset (Li et al., 2023) experiment, we follow the evaluation metrics defined by CHOIS (Li et al., 2024a):

**Condition Matching Metric.** Those metrics calculate the Euclidean distance between the predicted and input object waypoints. It includes the start and end position errors $(T_s, T_e)$, and waypoint errors $(T_{xy})$ measured in centimeters (cm).

**Human Motion Quality Metric.** Those metrics encompasses the foot sliding score (FS), foot height $(H_{feet})$, *Fréchet Inception Distance* $(FID)$ and *R-precision* $(R_{prec})$. FS is the weighted average of accumulated translation in the xy plane, following prior work He et al. (2022), measured in centimeters (cm). $H_{feet}$ assesses the height of the feet, also in centimeters. $R_{prec}$ and $FID$ are computed following the text-to-motion task Guo et al. (2022a). $R_{prec}$ (top-3) measures whether the generated motion is consistent with the text. $FID$ assesses the motion quality by computing the discrepancy between the distributions of ground truth and generated motions.

**Interaction Quality Metrics.** Those metrics assess the accuracy of hand-object interactions, encompassing both contacts and penetrations. For contact accuracy, it employs precision $(C_{prec})$, recall $(C_{rec})$, and F1 score $(C_{F_1})$ metrics following prior work Li et al. (2023). Additionally, it includes contact percentage $(C_{\%})$, determined by the proportion of frames where contact is detected. To compute the penetration score $(P_{hand})$, each vertex of the hand $V_i$ is used to query the precomputed object's Signed Distance Field (SDF). This process yields a corresponding distance value $d_i$ for each vertex. The penetration score is then derived by computing the average of the negative distance values (representing penetration), formalized as $\frac{1}{n}\sum_{i=1}^{n}|min(d_i, 0)|$, measured in centimeters (cm).

We note that CHOIS additionally measured the distance of the generated motion from the corresponding ground truth motion. Since HOIDiNi is a generative model, not aiming to reconstruct the ground truth, we find this metric irrelevant for our scope and omit it.

# E  USER STUDY

We conducted a user study for the GRAB dataset (Taheri et al., 2020a) with 24 participants, evaluating, in total, 12 side-by-side randomly selected samples of two models using the same inputs. We asked the user to evaluate the *grasp quality*, *prompt adherence*, and *overall quality*. As shown in Figure 6, users preferred the results generated by our framework. A representative screenshot from the study interface is shown in Figure 8.

# F  LLM USAGE

In this paper, we used ChatGPT 4o/5 to revise our writing, code assisting, and to enhance the GRAB (Taheri et al., 2020a) labels into text prompts.

