# OpenReview forum: "HOIDiNi: Human-Object Interaction through Diffusion Noise Optimization"
_ICLR.cc/2026/Conference — ICLR 2026 Conference Withdrawn Submission_

### Official Review · Reviewer_yxAs · 2025-10-30

**Soundness:** 3
**Presentation:** 2
**Contribution:** 3
**Rating:** 2
**Confidence:** 5

**Summary:**

This paper proposes a new method for text-driven HOI generation, which consists of two phases. In the first object-centric phase, an object's 6DoF pose and contact labels on both the object's surface and hands are generated using a diffusion model. In the second human-centric phase, whole body motion is generated. Various loss terms are defined to ensure the coherence between the human and object motion. Quantitative and qualitative experimental results are reported on GRAB and OMOMO.

**Strengths:**

1. The proposed two-phase HOI generation framework is appealing and effective. It disentangles the HOI generation tasks, which involves synthesis of both human and object motion as well as ensuring their coherent interactions, into simpler tasks.

2. The visual results of generated HOIs are of high quality. Although there are still artifacts, the quality is generally higher than other baseline models.

3. The code and model checkpoints is promised to be released.

**Weaknesses:**

1. Using DNO to improve the contact between hands and objects is a critical component and contribution in this paper. However, except for the brief introduction of how DNO works in general in Section 3, it is never explained in detail how it is integrated into HOI. For instance, for the equation defined around line #187, what does $x_T$ correspond to in HOI generation? What is the definition of $R_{decor}$? Furthermore, DNO itself is not entirely novel. It is expected to see insights why using it is for the HOI generation task is a better choice than, say classifier guidance. Unfortunately, all of these are completely missing in the paper and appendix. The presentation of the paper needs significant revision

2. The proposed approach is restricted to work for handling the contact of hands and objects only. While hands-object interactions generation itself is also challenging, such a limitation should be explicitly acknowledged in the paper.

**Questions:**

1. Is the same diffusion model (CPHOI diffusion) used in both phase 1 and 2?

2. In Table 1, how does the "Classifier Guidance" variant work?

3. How are the loss terms $L_{Object}$ and $L_{pen}$ combined?

4. As shown in Fig. 8 in Appendix, only skeletons are shown for user study. Why not using the full mesh so users can better gauge quality, say by spotting the penetration artifacts?

5. In Table 2, for the condition matching part, all the numbers of HOIDiNi are zeros. How to interpret them? I didn't find any visual results for this part in the main paper, appendix, nor the supplementary video.

---

### Official Review · Reviewer_YGQ8 · 2025-10-31

**Soundness:** 2
**Presentation:** 3
**Contribution:** 2
**Rating:** 4
**Confidence:** 5

**Summary:**

The paper proposes HOIDiNi, a text-driven diffusion-based framework for synthesizing realistic and physically plausible human–object interactions (HOI). The key innovation is the introduction of Diffusion Noise Optimization (DNO), which optimizes in the noise space of a pretrained diffusion model, enabling fine-grained contact accuracy while maintaining motion realism. The method adopts Object-Centric Phase  model to predict object trajectories and dynamic hand–object contact pairs, and then use Human-Centric Phase model to refine full-body motion conditioned on the fixed object motion and contacts, ensuring physically correct yet natural movements.

**Strengths:**

- Separating the optimization into two phases (object -> human) significantly improves stability and realism, effectively addressing a key limitation of prior single-step methods.

- The model explicitly predicts semantic contact pairs rather than relying on heuristic nearest-neighbor matching, resulting in more consistent and physically meaningful grasps.

- The quantitative and qualitative results for second phase (whole body generation ) are both strong; in particular, from videos and images in the paper, the qualitative outcomes show smooth, realistic motions with no noticeable artifacts or penetrations.

**Weaknesses:**

- The paper primarily employs two datasets, but to my knowledge, OMOMO does't include hand motions. Therefore, what is the purpose of using Contact-Paris here? Additionally, I did not observe any visualization results on OMOMO.
- There are no quantitative or qualitative evaluations provided for the first (object-centric) phase to assess the quality of the generated object trajectories, accuracy of contact prediction and penetration score.
- The motion diversity and scale of the GRAB dataset are still relatively limited. It would strengthen the paper if the model were evaluated on larger, whole-body HOI datasets such as HUMOTO or InterAct et al.

**Questions:**

Please see weakness.

---

### Official Review · Reviewer_cv8a · 2025-10-31

**Soundness:** 3
**Presentation:** 2
**Contribution:** 2
**Rating:** 4
**Confidence:** 4

**Summary:**

Authors proposed a full-body HOI motion generation pipeline which can generate it from text prompts and object geometry. Authors employed CPHOI that jointly models (1) the contact between hand anchors and object surface points, (2) full human motions and (3) object trajectories. To prevent unstable joint optimization, authors split the generation into two phases, which are object-centric and human-centric phases. Experiments involve GRAB and OMOMO datasets and demonstrate superior performance compared to baselines such as IMoS and CHOSIS.

**Strengths:**

Good motivation: The trade-off between realism and accuracy is the issue in previous methods and needs to be tackled to achieve the successful HOI motion generation.

Some ideas are novel: Dividing the optimization into human-centric and object-centric phases seems like an interesting idea.

**Weaknesses:**

- Some important references are missing while comparing their method only with two baselines (ie. IMoS and CHOSIS). Authors need to include them in the text. Also, I think authors need to empirically compare their baseline with ChainHOI.

ChainHOI: Joint-based Kinematic Chain Modeling for Human-Object Interaction Generation, CVPR’25.
Himo: A new benchmark for full-body humans interacting with multiple objects, ECCV’24.

- Furthermore, for hand-only scenario, below references are missing:

Text2HOI: Text-guided 3D Motion Generation for Hand-Object Interaction, CVPR’24.
DiffH2O: Diffusion-Based Synthesis of Hand-Object Interactions from Textual Descriptions, SIGGRAPH’24.

- Two-phase pipeline is empirically proven effective while their theoretical justification is not presented.

- Ablation study could be more complete: I think the effectiveness of the DNO scheme also needs to be validated, by showing the performance with and without the scheme.

- Unseen object generalization is also an important aspect while hasn’t been considered.

**Questions:**

Time complexity is not discussed. How long does DNO take per sample for each phase?

How robust is the proposed pipeline to unusual object geometries or very small, thin objects?

Could the two-phase scheme be inverted (human-first, then object-follow)?

---

### Official Review · Reviewer_jhp5 · 2025-11-01

**Soundness:** 3
**Presentation:** 3
**Contribution:** 3
**Rating:** 4
**Confidence:** 4

**Summary:**

1. This paper employs Diffusion Noise Optimization (DNO) to achieve accurate contact modeling and realistic motion generation.

2. It aims to preserve the diffusion noise within the manifold of realistic human motion.

3. Since applying DNO jointly to contact, human, and object components is highly challenging, the optimization is divided into two separate phases.

4. The optimization process consists of two phases: an object-centric phase and a human-centric phase. In the object-centric phase, diffusion noises are optimized for contact points and object motion. In the human-centric phase, the generated contact and object motion are used as conditions to synthesize realistic human motion.

**Strengths:**

1. The two-phase approach is well-motivated, as it first generates object-centric motion and then fixes and conditions it to produce human-centric motion.

2. It predicts human–object contact for each frame, unlike existing methods that rely on heuristics.

3. The results appear realistic in the visual comparisons, clearly outperforming other baselines.

4. The graph in Figure 7 is impressive and supports the validity of the two-phase approach.

5. The condition-matching results in Table 2 achieve perfect performance.

**Weaknesses:**

1. Figure 1 looks visually appealing but lacks essential information, such as the weaknesses of previous work, specifically, the trade-off between realistic motion and accurate contact modeling.

2. In Table 1, the penetration metric performs worse than IMOS, likely because IMOS generates motions with fewer contact instances, which also reduces motion diversity. What do you think of it?

3. There is semantic misalignment between the object and the mouth in the generated motion for the prompt “drinking from a bottle”, as shown in the supplementary video.

4. The paper provides very limited comparisons against existing methods. [1], [2].

5. It would be beneficial to visualize prediction results on the OMOMO dataset, comparing with Human-Object Interaction from Human-Level Instructions [2], whose results appear realistic and physically plausible.

6. On the GRAB dataset, Text2HOI (hand–object interaction) [3], an inference-only method, can also generate realistic-looking motions. This suggests that the key factor lies more in the dataset than in the method itself. Thus, I wonder the proposed method can generate realistic and plausible motions on the OMOMO dataset as well.

[1] Peng, Xiaogang, et al. “HOI-Diff: Text-Driven Synthesis of 3D Human-Object Interactions Using Diffusion Models.” Proceedings of the IEEE/CVF Conference on Computer Vision and Pattern Recognition, 2025.
[2] Wu, Zhen, et al. “Human-Object Interaction from Human-Level Instructions.” Proceedings of the IEEE/CVF International Conference on Computer Vision, 2025.
[3] Cha, Junuk, et al. “Text2HOI: Text-Guided 3D Motion Generation for Hand-Object Interaction.” Proceedings of the IEEE/CVF Conference on Computer Vision and Pattern Recognition, 2024.

**Questions:**

1. What is jitter loss?

2. Table 1 needs some edits, especially in the AVE metric. IMoS and Ours achieve the same value (0.121), but only Ours is bold-highlighted. Also, Phase 1 (Inference) and Phase 2 (DNO) achieve the highest values, but since a lower AVE indicates better performance, this should be corrected.

3. Please check the Diversity and Multimodality metrics in Table 1 as well. You noted that the right arrow indicates that better performance is closer to the GT value. However, in Diversity, Ours is closer to GT than IMoS, while in Multimodality, IMoS is closer to GT.

---

### Official Review · Reviewer_81dR · 2025-11-01

**Soundness:** 2
**Presentation:** 3
**Contribution:** 2
**Rating:** 4
**Confidence:** 4

**Summary:**

The paper introduces HOIDiNi, a novel text-driven diffusion framework designed for synthesizing realistic and plausible human-object interactions (HOIs). The framework addresses the challenges of achieving both realistic motions and accurate contacts in HOI generation by optimizing directly in the noise space of a pretrained diffusion model using Diffusion Noise Optimization (DNO). HOIDiNi separates the optimization process into two phases: an object-centric phase that determines hand-object contact locations with object trajectories and a human-centric phase that refines the full-body motion. This two-phase  approach allows for precise hand-object contact without compromising motion naturalness. Quantitative, qualitative, and subjective evaluations on the GRAB and OMOMO datasets demonstrate that HOIDiNi outperforms prior works and baselines (IMos, Chois) in contact accuracy, visual plausibility, and overall quality. Video demonstrates fine-grained hand-object maniplation under full-body motion.

**Strengths:**

1. The video has demonstrated high-quality hand-object maniplation motions. And a clear improvement over Imos and chois for hand-object grasping is shown.

2. HOIDiNi is guided by textual prompts, making it versatile for various applications where user input is essential.

3. A user study indicates a strong preference for the motions generated by HOIDiNi over competing methods.

**Weaknesses:**

1.	As the paper cites bimArt, a paper with the same idea should also be cited “ManiDext: Hand-Object Manipulation Synthesis via Continuous Correspondence Embeddings and Residual-Guided Diffusion, TPAMI 2025”.

2.	As the main results are hand-object grasping, it would be better to compare with some hand-only object manipulation papers for the quality of hand-object contact.

3.	The paper mainly compares IMos and Chois. However, although the paper is titled “human-object interaction”, the video only demonstrates results of manipulating on-desk objects by hands, without showing more interactions about larger objects in a large room, such as “lift the chair”, “moving the trashcan” just like chois do. This makes me feel that the title is an overclaim. This paper is more like a “hand-object interaction under full-body motion scenarios”.

4.	The pipeline is too straight-forward, as the second phase is quite similar to bimArt and ManiDext—given the object motion, generate hand motions. Therefore, I question about the effectiveness of the hand-object interaction representation compared with bimArt or ManiDext.  As one of the key contributions, a more detailed analysis would strengthen the paper. E.g. how the anchor number affect the contact prediction performance/final performance? How was the object scaled for contact location learning?

5.	The method may face generalization issue to a wide range of objects and interactions beyond those covered in the training data.

**Questions:**

See above.

---

### Note · Authors · 2025-11-27

I have read and agree with the venue's withdrawal policy on behalf of myself and my co-authors.